# Dialysis Reimbursement: What Impact Do Different Models Have on Clinical Choices?

**DOI:** 10.3390/jcm8020276

**Published:** 2019-02-25

**Authors:** Giorgina Barbara Piccoli, Gianfranca Cabiddu, Conrad Breuer, Christelle Jadeau, Angelo Testa, Giuliano Brunori

**Affiliations:** 1Department of Clinical and Biological Sciences, University of Torino Italy, 10100 Torino, Italy; 2Nephrologie, Centre Hospitalier Le Mans, 72000 Le Mans, France; 3Nephrology, Brotzu Hospital, 09100 Cagliari, Italy; gianfranca.cabiddu@tin.it; 4Direction, Centre Hospitalier Le Mans, 72000 Le Mans, France; cbreuer@ch-lemans.fr; 5Centre de Recherche Clinique, Centre Hospitalier Le Mans, 72000 Le Mans, France; cjadeau@ch-lemans.fr; 6Association ECHO, 44000 Nantes, France; atesta@echo-sante.com; 7Nefrologia, Ospedale di Trento, 38100 Trento, Italy; gcbrunori@hotmail.com

**Keywords:** dialysis reimbursement, costs, renal replacement therapy, incremental dialysis, predialysis care

## Abstract

Allowing patients to live for decades without the function of a vital organ is a medical miracle, but one that is not without cost both in terms of morbidity and quality of life and in economic terms. Renal replacement therapy (RRT) consumes between 2% and 5% of the overall health care expenditure in countries where dialysis is available without restrictions. While transplantation is the preferred treatment in patients without contraindications, old age and comorbidity limit its indications, and low organ availability may result in long waiting times. As a consequence, 30–70% of the patients depend on dialysis, which remains the main determinant of the cost of RRT. Costs of dialysis are differently defined, and its reimbursement follows different rules. There are three main ways of establishing dialysis reimbursement. The first involves dividing dialysis into a series of elements and reimbursing each one separately (dialysis itself, medications, drugs, transportation, hospitalisation, etc.). The second, known as the capitation system, consists of merging these elements in a per capita reimbursement, while the third, usually called the bundle system, entails identifying a core of procedures intrinsically linked to treatment (e.g., dialysis sessions, tests, intradialyitc drugs). Each one has advantages and drawbacks, and impacts differently on the organization and delivery of care: payment per session may favour fragmentation and make a global appraisal difficult; a correct capitation system needs a careful correction for comorbidity, and may exacerbate competition between public and private settings, the latter aiming at selecting the least complex cases; a bundle system, in which the main elements linked to the dialysis sessions are considered together, may be a good compromise but risks penalising complex patients, and requires a rapid adaptation to treatment changes. Retarding dialysis is a clinical and economical goal, but the incentives for predialysis care are not established and its development may be unfavourable for the provider. A closer cooperation between policymakers, economists and nephrologists is needed to ensure a high quality of dialysis care.

## 1. Introduction

Renal replacement therapy (RRT) is a life-saving, long-lasting, expensive treatment. In Europe, Japan, the United States and Canada, about one person in 1000 is presently alive thanks to dialysis or transplantation [1,2,3]. The prevalence is lower in most emerging countries, due to low or incomplete availability of dialysis and transplantation, lower diagnostic yield, in particular in remote or rural areas, and the presence of competitive mortality, specifically from infectious and cardiovascular illnesses, whose effect is to limit the number of patients who live long enough to reach end-stage renal disease (ESRD) [2,3,4].

Overall, it is estimated that the need for dialysis will rise sharply in developing countries, reaching even higher levels than in the Western world. The case of Taiwan, where, for reasons that still are not fully understood, the prevalence of end-stage renal disease is as high as 3000 people per million, is emblematic [5,6].

The situation in high-income countries is equally complex. While the incidence of ESRD patients needing RRT has apparently reached a plateau [2,3], it is, however, difficult to quantify the role of recent changes in the indications for dialysis [7,8,9,10,11,12,13]. The policy of early dialysis start has not, as was hoped, increased rates of survival.

Conversely, a great deal of recent evidence suggests that an “intent to defer” policy is preferable, acknowledging the advantage of retarding the start of RRT in clinically stable patients, instead of determining “by numbers” when dialysis should start, i.e., beginning dialysis when the estimated glomerular filtration rate (eGFR) drops below a pre-defined level [7,8,9].

Furthermore, RRT is not without adverse effects, in particular on the patient’s quality of life, and the negative effects are more serious in elderly individuals and in patients with high comorbidity. In these patients, the advantages in terms of survival are relatively low, which has led to increasing the number of elderly patients managed with “palliative” or “conservative” therapy [7,8,9,10,11,12,13].

Due to the combination of high costs and widespread need, renal replacement therapy is one of the most expensive treatments where it is available without restrictions [14,15,16,17,18]. Commonly cited figures report that RRT consumes between 2% and 5% of the global healthcare budget, an estimate that is worrying in the context of the global economic crisis [14,15,16,17,18].

The two main RRT modalities, dialysis (in both its forms: peritoneal dialysis (PD) and extracorporeal dialysis (hemodialysis, HD)) and kidney transplantation, have different costs and different results: kidney transplantation is the best form of RRT, both clinically and economically, provided that no contraindication is present and the patient accepts the need for life-long immunosuppressive treatments [19,20,21,22,23,24,25,26]. While the new approaches are presently widening the indications, severe and diffuse cardiovascular diseases, neoplasia, some chronic infectious diseases, and other conditions such as drug addiction, morbid obesity, and recurring primary kidney diseases on previous transplants continue to limit the indications for kidney transplantation [20,21,22,23,24,25]. Age is a further limiting factor; age is often associated with comorbidity and the prevalence of patients who can be waitlisted obviously decreases with age. Besides comorbidity, although the age limit has progressively risen and has now reached 80 in many Western European countries, tighter limits are usually present in developing countries or in settings where the availability of organs for transplantation from cadaveric and living donors is limited [27,28,29,30,31,32,33]. As a result, the prevalence of potential candidates for kidney transplantation ranges from 30% to 70% of the overall RRT population; even in the most favourable cases, access to kidney transplantation is not always immediate and, even with a wise combination of resources (cadaveric and living kidney donors, preemptive kidney transplantation), waiting lists are long and most patients experience a period of dialysis [32,34,35,36,37].

The cost of dialysis is therefore one of the main factors still determining overall RRT expenditure; furthermore, due to selection criteria, the most complex patients remain on dialysis, increasing the overall cost of care.

While its economic importance is clear, the actual costs of dialysis are not uniformly defined and the figures obtained in different countries are not easy to compare [38,39,40,41,42]. This is due to different costs for the same items, different ways of organizing care, and the inclusion or exclusion of key items (e.g., blood tests, drugs, transportation) in the “dialysis budget” [38,41]. The distribution of costs differs significantly: in highly resourced countries the most important item is the cost of nursing and medical care. Conversely, the cost of dialysis supplies is the most relevant item in most emerging countries. A good marker of the differences is the reuse of dialysers: in highly resourced countries, the cost of working time needed for processing is too high to make this procedure cost-effective, while the reverse is true in many emerging countries where the reuse of dialysers is still a common practice [43,44,45].

Reimbursement for dialysis follows different rules worldwide, including or excluding some items, and considering quality requirements or not. Large studies, conducted to analyse differences in dialysis policies, and investigate what impact different policies have on the results of treatment, like the DOPPS (dialysis outcomes and practice pattern) study, have made the medical community aware that while the care of the individual patient is important, how the system is organized is also a key factor [46,47,48,49,50].

Meanwhile, dialysis is undergoing a series of fundamental clinical changes.

In common with most other branches of clinical medicine dealing with chronic diseases, the shift from standardization to personalization has had an impact on perspectives and care [51,52,53].

The present opinion paper has been planned to discuss the potential advantages and drawbacks of different policies of reimbursement for dialysis. Taking into account the development of personalized treatments, it focuses on four paradigmatic issues: the relationship between haemodialysis and peritoneal dialysis; incremental dialysis; intensive haemodialysis; and predialysis care. The authors have used their countries, Italy and France, as main examples of how a given healthcare system can have an impact on the overall care of kidney patients.

## 2. Costs and Reimbursements: Not the Same Story

Expenditure for dialysis and reimbursement for dialysis are closely linked and mutually influence each other. However, they do not have the same meaning [14,15,41].

Costs depend on structure, organization, supplies, and healthcare personnel. Although they are also significantly influenced by social and political issues (e.g., the cost of healthcare workers depends on salaries), costs are largely determined by medical choices (organization of the dialysis ward, choice of materials etc.). The reimbursement system is usually determined by policy decisions (favouring in-hospital or out-of-hospital treatment; financing public or private structures; increasing high-tolerance modalities, etc.) [14,15,41,54,55,56,57,58].

For example, in Europe, Canada and the United States, as well as in Australia, the cost of healthcare professionals has a greater impact than the cost for materials; the costs of the “structure” (private or public hospitals, etc.) vary widely, and may be relevant in particular in countries where the efficiency of the healthcare system is low, as measured by the high “indirect” costs (costs of the overall hospital structure, including or excluding transportation) that are not always declared but may be as high as 20–30% of the overall expenditure [59]. Overall, in Europe, the costs of materials differ little, while there is a significant range of salaries, which is only partly compensated for by differences in workload: for example, a French centre with up to 15 dialysis beds employs at least two full-time nephrologists, and one with up to 30 dialysis beds at least three. This means, for example, that a pool of 80 in-hospital dialysis patients can be managed by only two full-time physicians, while out-of-hospital figures are even lower [60]. Italian figures are less well defined, but the current rule is that at least six nephrologists are needed in each nephrology structure, thus assuming that a higher number of specialists is needed in a medium-sized dialysis ward. This policy, originally intended as a way to ensure the presence of an adequate number of nephrologists in centres in small towns and rural areas, led to a decrease in the independence of small nephrology structures, and many of them were absorbed by larger internal medicine wards [61]. France and Italy have roughly the same resident population, but Italy has almost twice as many nephrologists as France. The higher number of nephrologists in Italy partially compensates for a lower number of secretaries, nurses and aides. The difference in salaries is difficult to assess, due to the high variability between public and private, and in Italy among regions.

This difference also has an important impact on research. Physicians working in French hospital centres are encouraged to conduct studies and publish articles on their research by the SIGAPS-SIGREC system (SIGAPS being for Système d’interrogation, de gestion et d’analyse des publications scientifiques, and SIGREC for Système d’information et de gestion de la recherche et des essais cliniques), which over a period of four years, starting from the year after publication, allocates about €64,000 for each paper published (first or last name) in a journal ranking in the first 10% in its field, and up to €8000 for a paper (first or last name) published in a journal ranking in the last quartile [62]. These incentives do not exist in other countries, such as Italy; however, a gross analysis of the Pubmed database for the year 2017, employing the terms dialysis, haemodialysis or haemodiafiltration and Italy or France, retrieved 665 papers in Italy and 404 in France. While the issue is complex, these data suggest that a higher number of specialists is more efficacious than a high, but delayed economic reward, and that the latter should probably be at least partially converted into employing a larger work force.

## 3. Dialysis Reimbursement: Per Session, Per Patient, Per Bundle

There are three main ways of calculating the cost of dialysis and establishing how it should be reimbursed. The first involves dividing dialysis into a series of elements and reimbursing each one separately (dialysis itself, medications, intradialytic drugs, transportation, chronic treatments, laboratory tests, imagery; consultations; hospitalizations; home assistance). The second, known as the capitation system, consists of merging these elements, partially or entirely, in a per capita reimbursement, while the third, usually called the bundle system, entails identifying a core of procedures intrinsically linked to treatment (e.g., dialysis sessions, tests, drugs and transportation).

Each one has advantages and drawbacks, and each one impacts differently on the organization and delivery of dialysis care, as will be discussed in the pages that follow.

## 4. Reimbursement Per Separate Element: Dialysis Treatment Seen as a Matryoshka

Delivering dialysis entails more than merely delivering a session of blood purification. Compensating for a lack of kidney function also includes the use of medications (from erythropoietin to anti-hypertensive drugs), controlling the efficacy of dialysis sessions via regular blood tests, and checking for cardiovascular diseases and other frequently associated comorbidities.

The first advantage of dealing with each item separately is that this allows us to better understand the cost of each one, targeting actions needed to control costs to specific issues, such as transportation or blood tests (Figure 1).

A second advantage is that the different items do not compete with one another, and this helps to protect clinical decisions from being influenced by global budget constraints (for example, transportation costs, higher in rural areas, do not compete with costs for blood tests in the same settings).

A third element in favour of separating items is that in a given setting the amount spent on a dialysis session (dialyser, dialysis machine, healthcare workers) is similar for all patents, while the costs of check-ups, drugs and imagery largely depend on age and comorbidity, and even in the same setting can vary widely from patient to patient. Thus, separating the elements may more easily allow for stratification and may help justifying cost differences, for example as for comorbidity. For instance, a four-hour haemodiafiltration session, performed with a high-flow membrane, has a supply and nursing cost that is roughly the same for a 40-year-old patient who started dialysis two years previously, is waitlisted for kidney transplantation and has a low comorbidity score, and for an 80-year-old patient, with high comorbidity and severe cardiovascular disease. However, the cost of drugs, biochemical controls and imagery increases with age and comorbidity, and a separate analysis is more likely to capture the differences.

The cons are, however, many. While this approach is appealing in the care of complex patients, since it avoids potentially dangerous interferences between the items and phases of care, it could lead to limits on the overall budget dedicated to the more complex patients. Separating the different items is generally difficult, and if the distinction corresponds to a separation of providers or payers (as it does in France, where transportation and in-hospital care have separate budgets), an overall advantage of one therapeutic choice may be missed, or result in a paradoxical disadvantage to one of the parties, as the case of incremental dialysis, discussed in a later paragraph, shows.

Furthermore, separating items may lead to a focus on issues of lesser relevance and forgetting others; one example, derived from the Italian experience, may be the emphasis put on the reduction of the cost and number of blood tests or consumables, completely forgetting the cost of dialysis waste management, which could be as high as 50% of the overall cost of a new dialyzer and blood lines [63]. In this regard, the separation of the items may lead to losing sight of the overall problem.

## 5. The “Capitation System” of Reimbursement: Dialysis Treatment Seen with a Distributive Approach

There is an obvious advantage to merging everything entailed in dialysis treatment into a single “mega” reimbursement payment [64]. Patients need integrated care, and integrating reimbursement supports a holistic view and helps to avoid fragmenting treatment (Figure 2). Furthermore, it can make it possible to reinvest in specific aspects of care by favouring the careful distribution of the overall budget. An example is home assistance for patients who wish to be treated at home but who lack a partner for dialysis. Lowering transportation and hospital costs means the money saved can be used to pay a helper, a system that has allowed peritoneal dialysis to be more widely used in some areas in Italy [65,66].

In addition, such a system makes it possible to bypass the need to define a maximum affordable cost per item per patient, thus allowing a nephrology centre to allocate more resources to fragile patients, whose costs are counterbalanced by those of younger and fitter patients, who are less clinically demanding. 

In such a context, physicians act as “resource regulators” whose role is to favour the use of the least expensive options for each item, and make money available to pay for more expensive treatments for special cases. An example is expanding home dialysis and investing in in-centre daily dialysis for fragile or pregnant patients. This i, however, not fully the case in the United States, where a capitation system was recently modified towards a bundled care system, with positive effects on the development of home care [67,68].

There are two requisites for the smooth functioning of the capitation system: dealing with a critical mass of patients, and treating patients with a different case mix (Figure 2). In other words, performance is optimal only when a sufficient number of patients are treated (more than would normally be in care in a small dialysis centre) to allow physicians to reallocate resources.

Furthermore, due to the obvious attrition that accompanies kidney transplantation (or with out-of-hospital dialysis, especially when managed by different providers), the case mix may be uniformly high in in-hospital centres. The paradoxical risk is to penalize the centres with the best overall performance (high and rapid access to kidney transplantation; wide use of out-of-hospital dialysis). The rate of attrition may be particularly important when the system is mixed, for-profit and non-profit, since for-profit structures will tend to select the “least complicated” and therefore least expensive patients [69,70]. A strict capitation system may therefore induce a selection process that is potentially detrimental for non-profit structures, which are, on the contrary, those that tend to have better results [69,70].

Correction for comorbidity can partially correct for these discrepancies. However, assessment of comorbidity is complex; no system is uniformly the best one and the definition of frailty, nutritional status and comorbidity is either very subjective, and not graded, or very complex (and never devoid of a subjective component) [71,72,73,74,75,76,77,78,79,80].

## 6. What Is Favourable for the Patient and for the System May Not Be Favourable for the Hospital: The Case of Incremental Dialysis

In incremental dialysis, patients start treatment with one or two sessions per week and progressively increase to a full dialysis schedule, or even daily dialysis [52,81,82,83].

The cost of the supplies for each session does not change, while the cost per patient, for example per month, is deeply affected by the clinical choice of 1–2 (incremental) or 4–6 (intensive) dialysis sessions per week. The usual policy is to check the results in incremental dialysis more frequently than in conventional dialysis. Therefore, if expenditures for blood tests have to be added to a payment per session, the single session per week costs more, while the total treatment cost is lower. Furthermore, managing patients with personalized treatments makes organizing the dialysis ward more complex.

Using a system of incremental dialysis, the same number of patients can be treated in a lower number of sessions. This means that, seen in the context of reimbursement per session of treatment, incremental dialysis is advantageous for the payer (the healthcare system in France and Italy: fewer sessions, less spent for transportation), but leads to higher expenditures for the provider (public hospital or private provider: difficult and time-consuming organization of the occupation of dialysis posts; higher cost of check-ups where they are considered as part of what a dialysis session costs). Since it is usually the provider that has the final say on the matter, the obvious risk is to disincentivize options that allow a centre to provide better, more personalized care, since they are more complicated and less lucrative.

The reverse would be true for a capitation system, where fewer global resources are employed for patients on less frequent dialysis, with a potential advantage for the provider, but with the risk of keeping the number of dialysis sessions to a low, unsafe level.

## 7. Is Bundled Care the Solution? Defining the Core, Defining Comorbidity: A Difficult Mission

An appealing alternative would be to identify a core of dialysis-related activities so that these could be reimbursed together, plus a series of specific “frequent activities” that would be reimbursed according to need (Figure 3). This is what is called the bundled system of care, also referred to as episode-based payment, episode payment, case rate, evidence-based case rate, global bundled payment, and package pricing [40,67,84,85,86,87,88,89]. Intended to be a middle way between the fee-for-service payment and capitation, this system would determine the amount of reimbursement due on the basis of expected costs for clinically defined “episodes” of care. The concept is appealing and is already partially integrated in the reimbursement of dialysis in many European countries (for example, erythropoiesis-stimulating agents (ESAs) are included in most fees for dialysis sessions). The effect of such a shift in payment policy is enormous. For example, the studies dealing with changes in the use of ESAs in the USA highlight how inclusion in the bundle changed clinical practice, with an enormous reduction in the use of ESAs in favour of higher iron levels. It remains to be determined whether this improved, impaired or had no effect on survival results. Yet, regardless of results, ESAs are a good example of how ethics and economics are linked and demonstrate that medical practice can be rapidly affected by changes in reimbursement policies [86,87,88,89]. The ESA experience shows the need for a careful analysis of the potential effects of further changes in the bundled payment system, for example with the inclusion of oral drugs, initially foreseen for 2025.

The potential advantage of the bundle is its flexibility. It can be designed differently, and is adaptable to a variety of contexts; however, unlike the capitation system, bundled reimbursement does not capture all costs, and differently from the fee-for-service model, it may make it difficult to disentangle what was spent on specific elements in the course of treatment. A well-designed bundle system should help clinicians to wisely meet their patients’ needs without discontenting providers, but often this is not what happens, and it is not easy to change the system so that it takes variations in patients’ care into account.

Once more, correction for comorbidity is possible, but there is no single score that precisely captures dialysis-related comorbidity, and given its complexity, variation over time, and the subjectivity of evaluation, grading comorbidity is usually not feasible [90,91,92,93,94].

## 8. A Fundamental Question: Haemodialysis or Peritoneal Dialysis?

The diffusion of peritoneal dialysis (PD) differs from country to country. The treatment is widely used in both rich and poor settings, in Canada, Australia and New Zealand, where distances make home treatment preferable, as well as in Mexico and Taiwan, where less expensive treatment options are chosen because of budgetary constraints [95,96,97,98,99,100,101,102].

Cost issues are, however, not limited to the emerging countries, since the weight of dialysis is remarkable in all contexts, and the increase in home treatments, and in particular in home haemodialysis, is advocated as a means to optimize costs and resources, with clinical outcomes at least equivalent to hospital dialysis [99,100,101,102].

Even if “peritoneal dialysis first” or “home haemodialysis first” probably represents a winning strategy for patients (more autonomy, more empowerment, better care), and for the health care system (lower costs of transportation, lower overall indirect costs and probably also lower costs of direct treatment, especially where PD is non assisted), this strategy is not uniformly developed, partly because of the fact that reimbursement is often lower and the advantage to the individuals and to society is not uniformly accompanied by an advantage to the dialysis providers [103,104,105,106,107].

Political decisions can play an important role: for example, the recent increase in peritoneal dialysis in Switzerland is due to a combination of favourable reimbursement for PD and a reduction in the reimbursement for haemodialysis if a minimum number of PD patients is not reached [98,99].

The availability of assisted peritoneal dialysis programs could profoundly change the penetration of peritoneal dialysis, in particular in elderly patients. However, the lower prevalence of PD in France, where assisted PD is the rule, as compared to Italy, where assisted PD is not available, once more indicates that things are not as simple as they may seem, and that economic incentives and drawbacks are just some of the potential factors determining treatment choices [96,97,98,99,100,101,102].

## 9. One-Size-Fits-All or Tailor-Made Treatments?

The heterogeneity of dialysis patients is a crucial point. It has been raised in all international comparisons and extensively discussed in relationship to costs [14,15,16,47,48,49,50]. In an era of precision medicine, individualized treatment and holistic approaches, delivering a fixed dose of dialysis to all patients can be likened to using the same washing machine setting for cotton and cashmere (Figure 4).

Furthermore, some individuals, in particular if affected by multiple and severe comorbidity, may not gain any benefit from dialysis, in terms of morbidity and mortality; while the controversy about so-called “palliative” or “conservative” care is behind the scope of this review, the advantage of this open discussion is to point out that the need for dialysis cannot be reduced to a mere series of indexes, each of which is incomplete and potentially misleading [7,8,9,10,11,12,13,51,52,53,54,55,108,109,110,111,112,113,114].

The failure of early dialysis to prolong life and improve its quality has caused nephrologists to reflect on the negative effects of treatment [7,115,116,117,118,119,120]. This was also the starting point for reconsidering incremental dialysis and for realizing that, especially in elderly patients, the advantages of a high dialysis dose are often counterbalanced by the iatrogenicity of treatment [120,121,122,123,124,125]. Increasing the dialysis dose by increasing the number (and/or duration) of sessions may, conversely, be necessary in particular situations, such as pregnancy or high metabolic needs, or be a suitable way to attain tolerance in fragile individuals [49,50,51,52,53,125,126,127].

However, standardization is still the most commonly pursued policy, first because of its simplicity, secondly because it leaves an important part of dialysis management to nurses, thus reducing the number of physicians involved (and cutting costs), and finally because “working by numbers” may be culturally reassuring.

Personalization of dialysis is compatible with all reimbursement models, but can create problems in each of them: in a fee-for-service system, each session is reimbursed, and more frequent dialysis may be favourable for the provider; however, there may be limitations (for example, a maximum of three dialysis sessions per week are reimbursed, or only patients on three sessions per week are reimbursed), impairing flexibility and making treatment personalization difficult if not impossible.

In a capitation model, combining less frequent (incremental) and more frequent dialysis sessions allows for greater flexibility; once more, however, the model is not devoid of risks, in particular of limiting a higher number of dialysis sessions for economic advantages.

In a bundled system the “dialysis package” can be designed differently, allowing a certain degree of personalization (or not). The focus switches to the definition of the “package” itself, maintaining a balance between the need for flexibility and clear definitions.

## 10. Predialysis Care May Be Good for the Patient and for the Community, but Less Rewarding for the Hospital

Economic reasoning also applies to determining the policies of dialysis start. Retarding dialysis is a time-consuming task and the longer a kidney ailment progresses, the greater the need for clinical check-ups and blood tests. However, the average reimbursement for a clinical visit that will require at least 30 min of a physician’s time is 10% of what is paid for a dialysis session, which will normally entail no more than 5 min of medical controls.

Dialysis usually allows an economic advantage for the provider, once a critical mass of treatments is reached. This may not be the case for outpatient care.

The data about the “day hospital” in which patients are taken in for a one-day hospitalisation in the case of a need for complex diagnostics or treatments that cannot be performed outside of hospital, are likewise not reassuring; in France, it has been calculated that the overall cost in 2016 was over 800 euros (213 for logistics and “housing” and 227 for physicians and nurses), against a reimbursement of 614 euros per day.

The advantage for the patient and for society of safely retarding dialysis is intuitive, but there is hardly any advantage involved for the structure delivering predialysis care. This means that, while dialysis is expensive, it may be economically advantageous for the structure providing treatment.

Prevention is theoretically a good option in all its forms, even the latest ones (prevention of kidney disease should of course be the first goal; prevention of progression should be pursued in all chronic patients, but even in the last stage, stabilizing kidney disease may be seen as a form of “late” prevention of the need for dialysis start). Previous studies by our group suggested that delaying the start of dialysis by two years could save enough money to pay the salary of a nephrologist for a year. This crude estimate, intended to raise interest in secondary prevention of end-stage kidney disease, should be borne in mind in organizing nephrology care [128]. However, the budgets for predialysis and dialysis care are usually separate and it may be difficult to demonstrate that comprehensive care really helps retard dialysis start, an issue that arises in other contexts, for example the dietary management of chronic kidney disease [129,130,131,132,133].

There is a clear need for implementation of a comprehensive network of predialysis care to optimize resources; investment in medical care has the advantage of increasing the flexibility of nephrology structures and making more efficient use of physicians’ time. This could then be translated into time to dedicate to clinical tasks and research.

## 11. Concluding Remarks

In the best scenario, all patients in all countries would receive all the treatment they need to preserve life and its quality as long as possible. Personalization, integration and flexibility are increasingly included in this comprehensive vision. Since this is not the rule, but still a goal to pursue, experienced clinicians should probably spend more time with economists and policy-makers to ensure the wise use of our finite resources, and, in line with developments in medical knowledge, adapt our always-imperfect systems to patients’ changing needs.

## Figures and Tables

**Figure 1 jcm-08-00276-f001:**
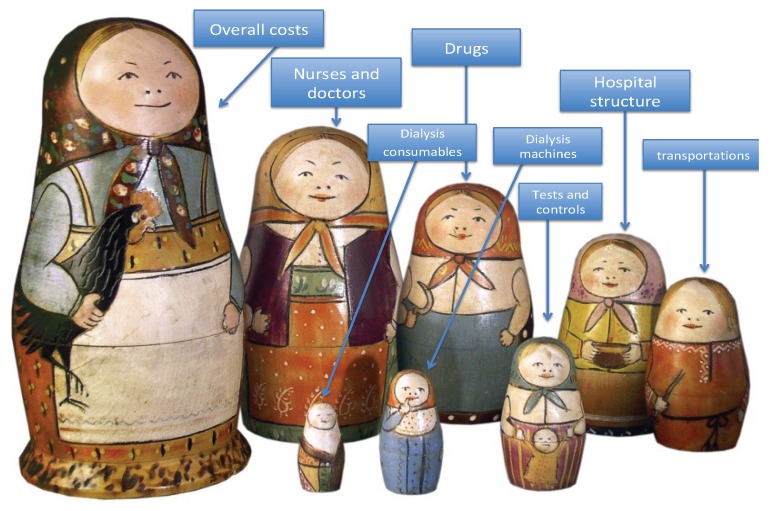
Dialysis costs as a matryoshka.

**Figure 2 jcm-08-00276-f002:**
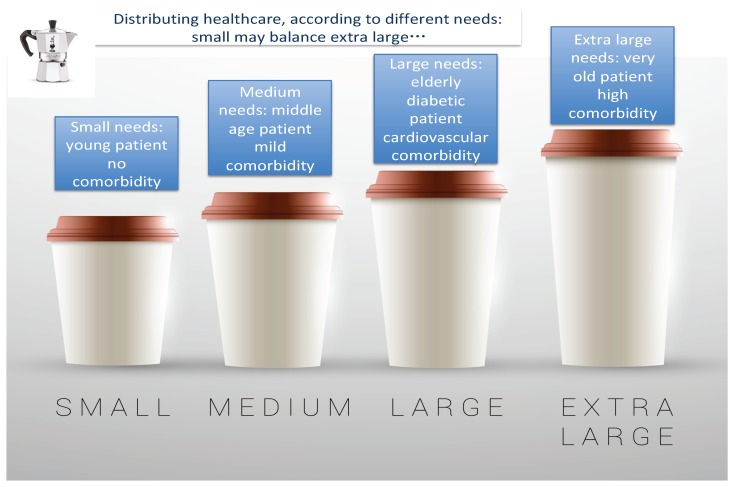
A distributive approach to dialysis reimbursement.

**Figure 3 jcm-08-00276-f003:**
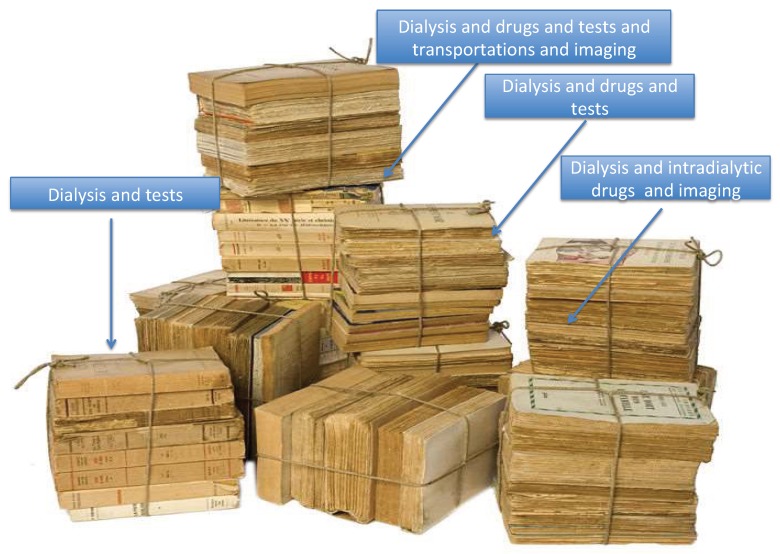
A bundle is not “just a bundle”; its meaning changes according to what is tied together.

**Figure 4 jcm-08-00276-f004:**
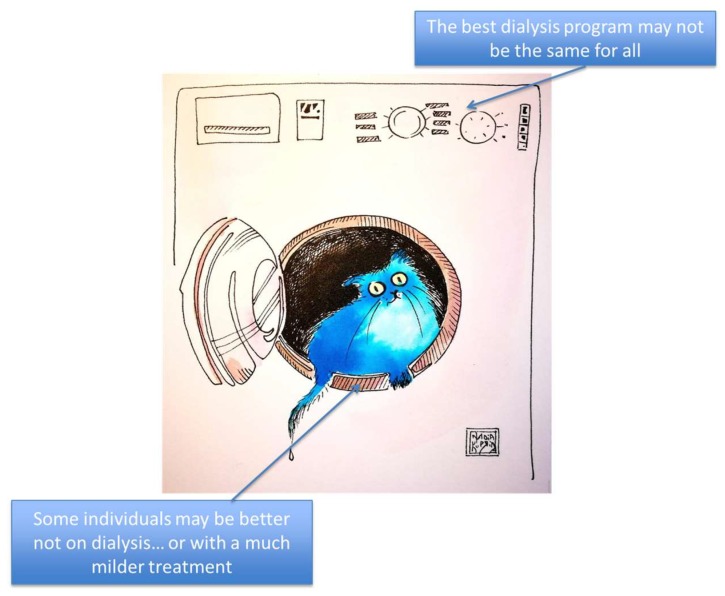
Different programs for the “washing machine”: choosing the “program” does not only regard tailoring treatments, but also the decision of whether or not to start renal replacement therapy.

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
