# Peer review of "Dialysis Reimbursement: What Impact Do Different Models Have on Clinical Choices?"

_jcm, 2019, doi:10.3390/jcm8020276_

Round 1
Reviewer 1 Report
Global evaluation
This interesting manuscript discusses different current dialysis reimbursement strategies in perspective of providers, government and patients. The authors clarify “fee for services” by visualizing a Matryoshka, “capitation system” by visualizing different size of coffee cups and “a bundle system” is unfortunately not visualized (can you give it a try?). The authors mention also their own experience (Italy and France) and many interesting examples are given. This manuscript appoints the ambiguity of the current European system, but it is not really clear in which country which strategy is provided, and what about the rest of world? Some parts are incomplete by lacking the pros and cons of each reimbursement strategy.
Unfortunately, not all strategies are discussed. The current reimbursement system is inadequate to increase the use of cost-saving dialysis (PD, home HD). Surprisingly, the authors did not mention ‘PD first’ strategy as an effective strategy to increase uptake of patients using this cost-saving modality, especially in wealthy countries.
The authors mention “delivery a fixed dose of dialysis to all patients, can be seen as using the same washing machine setting for coton and cashmere”: To my opinon, the authors better mention that cashmere should not be in the washing-machine. Patients with life threatening co-morbidities should rather be treated with conservative renal care than being on dialysis for their last life months. Initiating dialysis should be discouraged in these patients as dialysis is associated with worse outcome and low quality of life.
Comments:
Introduction-
5th alinea:
1) “the two main RRT modalities, dialysis (hemodialysis?) and kidney transplantation“ What do the authors mean with dialysis? Different modalities of dialysis are known with different costs and different repercussion on quality of life. PD and home HD can be cost saving and improve quality of life in patients compared to conventional HD.
2)the authors write in the introduction “ageing” is most important factor that limit the indication for kidney transplantation. Do you know something about the outcome of patients who were transplanted at an age between 70-80? This information is not given by your references. To my opinion these patients have a tremendous risk for infections, neoplasia and death. Thus transplantation can never replace dialysis for 100%.
Cost and reimbursement: not the same story:
3)3rd alinea: the authors mention “Europe, Canada and US”, but forgot to mention: Australia and New Zealand and maybe Singapore are wealthy countries too (high GDPc). What do you mean with “low efficiency of health care”; can you specify this?
4)4th alinea: is there difference between the average salary of Italian and France nephrologists? Can you discuss this.
Reimbursement per separate element.
5)5th or last alinea: this paragraph is totally unclear what do the authors mean with “potentially dangerous interferences”? the authors should also discuss the cons. One problem is that this strategy is very expensive (certainly not cost saving). Is there any literature that such system raises costs by innovation? (even when there is no proven benefit?). The authors should discuss that such a strategy does not stimulate cost saving dialysis modalities or consumenables.
A fundamental question: HD or PD?
6)First alinea: this alinea is not correct: Also in high income countries PD might be stimulated as cost saving treatment (Denmark) while PD in poor countries is mostly not cost saving as materials are more expensive than for HD.
7)3rd alinea: in this alinea “PD first” strategy should be discussed. In rich countries PD is cost saving and in poor countries the government has to reduce the import taxes of solutions or stimulates local manufacturing to reduce the cost of PD.
Pre-dialysis care may be good for the patient and for the community, but less rewarding for the hospital.
8)2nd and 3rd alinea should be deleted or replaced, this has nothing to do with retarding dialysis.
9)5th alinea: prevention and retarding dialysis is not the same issue. Prevention measures are taken to prevent that patients with CKD develop ESRD, while retarding dialysis is management of high phosphorus, potassium and metabolic acidosis with medication and low protein and potassium diet.
Reviewer 2 Report
This is an opinion paper; it scientifically sound conceptually and in its execution. The paper's aim is well justified in the light of subject matter importance across the developing economies and it has implications for the differences in the concept of 'cost of care' in kidney care (dialysis or transplantation), emerging care technologies, personalized medicine, reimbursement types, and other factors. The paper is timely and well written. There does not appear to be any particular issue to be corrected or modified for improvement. The policy conclusions flow logically from the analysis of the countries studied. Well done!
Author Response
the Authors would like to thank the reviewer for the kind appreciation of their study.
Round 2
Reviewer 1 Report
I am satisfied with the authors responses and have no additional comments.
Well done!